# Risk Prediction for the Development of Hyperuricemia: Model Development Using an Occupational Health Examination Dataset

**DOI:** 10.3390/ijerph20043411

**Published:** 2023-02-15

**Authors:** Ziwei Zheng, Zhikang Si, Xuelin Wang, Rui Meng, Hui Wang, Zekun Zhao, Haipeng Lu, Huan Wang, Yizhan Zheng, Jiaqi Hu, Runhui He, Yuanyu Chen, Yongzhong Yang, Xiaoming Li, Ling Xue, Jian Sun, Jianhui Wu

**Affiliations:** 1Key Laboratory of Coal Mine Health and Safety of Hebei Province, School of Public Health, North China University of Science and Technology, Tangshan 063210, China; 2College of Science, North China University of Science and Technology, Tangshan 063210, China; 3School of Public Health, North China University of Science and Technology, Tangshan 063210, China

**Keywords:** steelworkers, hyperuricemia, risk prediction

## Abstract

OBJECTIVE: Hyperuricemia has become the second most common metabolic disease in China after diabetes, and the disease burden is not optimistic. METHODS: We used the method of retrospective cohort studies, a baseline survey completed from January to September 2017, and a follow-up survey completed from March to September 2019. A group of 2992 steelworkers was used as the study population. Three models of Logistic regression, CNN, and XG Boost were established to predict HUA incidence in steelworkers, respectively. The predictive effects of the three models were evaluated in terms of discrimination, calibration, and clinical applicability. RESULTS: The training set results show that the accuracy of the Logistic regression, CNN, and XG Boost models was 84.4, 86.8, and 86.6, sensitivity was 68.4, 72.3, and 81.5, specificity was 82.0, 85.7, and 86.8, the area under the ROC curve was 0.734, 0.724, and 0.806, and Brier score was 0.121, 0.194, and 0.095, respectively. The XG Boost model effect evaluation index was better than the other two models, and similar results were obtained in the validation set. In terms of clinical applicability, the XG Boost model had higher clinical applicability than the Logistic regression and CNN models. CONCLUSION: The prediction effect of the XG Boost model was better than the CNN and Logistic regression models and was suitable for the prediction of HUA onset risk in steelworkers.

## 1. Introduction

Hyperuricemia (HUA) is a metabolic disorder disease that develops due to abnormal purine metabolism, resulting in elevated serum uric acid (SUA) concentrations [1]. A 2014 meta-analysis covering 16 provinces, municipalities, and autonomous regions in China showed that the prevalence of HUA in China was 13.3% (19.4% for men and 7.9% for women) [2]. Another meta-analysis in 2021, which included 2,277,712 subjects, showed that the prevalence of HUA had increased to 16.4% (20.4% for men and 9.8% for women) [3]. Previous studies have shown that the prevalence of HUA in China has doubled in the last 20 years and has become another public health problem of concern after diabetes [4]. Worldwide, the burden of gout has increased in 195 countries and regions, especially in developed countries and regions [5]. HUA is not only an early stage of gout but also an independent risk factor for coronary heart disease, hypertension, diabetes, and chronic kidney disease [6], which seriously endangers human health.

The steel industry is a pillar industry of the Chinese economy and directly employs as many as two million people. The health status of the workers is directly related to the development of the Chinese steel industry. It has been pointed out that steelworkers are exposed to occupational hazardous factors such as shift work, high temperature, and noise for a long time, and also have unhealthy habits such as smoking, alcohol consumption, and a high-salt diet, which cause or affect the risk factors of HUA differently from the general population [7]. Therefore, there is an urgent need to develop new risk prediction models for steelworkers’ morbidity, which can be used to improve the quality of life and health status of steelworkers.

Logistic regression is a traditional prediction model commonly used in the medical field and is widely used for a variety of disease predictions because of its clear parameter meaning and easy-to-understand outcome metrics. The convolutional neural network (CNN) is a feedforward neural network with a deep structure that is good at mining local features of data and extracting global training features and classification and has some advantages that traditional techniques do not have [8]. XG Boost, known as eXtreme gradient boosting, achieves classification by iterative computation of classifiers, and the addition of its regular term ensures the model’s robustness and reduces the time to process features because it was good at handling missing data [9]. We established the above three HUA morbidity risk prediction models based on the medical examination data information of more than two thousand steelworkers and compared their prediction effects, aiming to select the optimal model and provide a theoretical basis for the health management of this special occupational group.

At present, the popularization of HUA in China is still insufficient, the prevention and treatment situation is not optimistic, and the awareness rate and cure rate of HUA among patients are low [10,11]. Therefore, screening risk factors affecting HUA to establish prediction models, early identification, detection, and intervention of HUA patients has great social value to prevent and control the development of HUA and reduce the burden of the disease.

## 2. Materials and Methods

### 2.1. Study Design and Participants

The present study was a retrospective cohort study, relying on the Chinese National Key Research and Development Program “Beijing-Tianjin-Hebei Regional Occupational Population Health Effects Cohort Study”, which completed the baseline survey from January to September 2017 and the follow-up survey from March to September 2019. A total of 2992 steelworkers were included in the study, and the inclusion criteria for the study population were formal employees of the unit; more than 1 year of service; non-HUA patients at the time of the baseline survey; and voluntary signing of the informed consent form. Exclusion criteria were age > 60 years; and those with incomplete information. The study was reviewed and approved by the Ethics Committee of North China University of Technology (approval number: 16004).

### 2.2. Data Collection and Preprocessing

The subjects of this study were workers in the production department of Tangshan Iron and Steel Group who participated in the health examination, and all information was obtained from the baseline and follow-up surveys of the Beijing-Tianjin-Hebei cohort, including questionnaires, physical examinations, and laboratory examinations. The final data set was randomly divided into the training set (70% of observations) and the validation set (30%).

The questionnaire for the survey was developed by our team, one-on-one interviews were conducted by professionally trained PhD and MSc students from the School of Public Health of North China University of Technology to the workers of the steel enterprise.

Physical examinations were conducted by trained professional medical examiners according to standard testing methods for height, weight, blood pressure, and other indicators for workers in this enterprise.

For laboratory testing, fasting blood and morning urine were collected by the medical examination hospital before 9:00 a.m. daily and sent to the laboratory department of the medical examination hospital for uniform blood biochemical testing using a Myriad automatic biochemical analyzer (BS-800). The test indexes included fasting plasma glucose (FPG), uric acid (UA), total cholesterol (TC), triglyceride (TG), high-density lipoprotein cholesterol (HDL), triglyceride (TG), lipoprotein cholesterol (HDL-C), low-density lipoprotein cholesterol (LDL-C), creatinine (Cr), urea nitrogen (BUN), etc.

### 2.3. Definition of HUA

According to the Practice Guidelines for the Diagnosis and Management of Hyperuricemia in Renal Diseases in China (2017 edition) [12], developed by the Nephrologist Branch of the Chinese Physicians Association, blood uric acid ≥ 420 μmol/L in men and ≥360 μmol/L in women are being treated for HUA during the follow-up survey.

### 2.4. Definition of Variables

Hypertension: According to the classification criteria of the Chinese Guidelines for the Prevention and Treatment of Hypertension 2018 Revision [13], systolic blood pressure ≥ 140 mmHg and/or diastolic blood pressure ≥ 90 mmHg, or a previous history of hypertension and current use of antihypertensive drugs, were defined as hypertension;Diabetes: According to the classification criteria for glucose metabolic status in the Chinese guidelines for the prevention and treatment of type 2 diabetes mellitus (2020 edition) [14], fasting blood glucose ≥ 7.0 mmol/L, or a previous history of diabetes currently undergoing diabetes treatment was defined as diabetes mellitus;Smoking index: SI = number of cigarettes smoked per day × number of years of smoking [15]. The current study divided the smoking index into 3 groups according to the median: group 0 (0), group 1 (1~), and group 3 (300 and above);Drinking index: DI = years of drinking × (amount of liquor/month + 0.ll × amount of beer/month) [15]. The current study divided the drinking index into 3 groups according to the median: group 0 (0), group 1 (1~), and group 3 (1028.57 and above);The way of defining cumulative noise exposure, cumulative dust exposure, cumulative heat exposure, and cumulative days of night shift was detailed in the published article of the subject group [16];Physical exercise: more than three times a week, no less than 30 min each time;Body mass index: BMI = weight (kg)/height^2^ (m^2^). The normal range of body weight was BMI < 24 kg/m^2^, the overweight range was 24.0 kg/m^2^ ≤ BMI < 28.0 kg/m^2^, and the obese range was BMI ≥ 28.0 kg/m^2^;Dyslipidemia: according to the Chinese guidelines for the prevention and treatment of dyslipidemia in adults (revised version 2016) [17], serum total cholesterol ≥ 6.2 mmol/L, and/or triglycerides ≥ 2.3 mmol/L, and/or LDL cholesterol ≥ 4.1 mmol/L, and/or HDL cholesterol < 1.0 mmol/L, a previous history of hyperlipidemia, or the current use of lipid-lowering drugs was defined as dyslipidemia;Physical activity: The physical activity of workers was investigated using the International Physical Activity Questionnaire (IPAQ) (long-volume version) [18], and an overall weekly force activity level < 600 MET-min/w was defined as low-intensity operations, an overall weekly force activity level ≥ 600 MET-min/w was defined as medium-intensity operations, and weekly overall force activity level ≥ 3000 MET-min/w was defined as high-intensity operations;Occupational tension: The revised Chinese Work Content Questionnaire [19] (JCQ) was used to assess occupational stress, using the ratio of job requirements to degree of work autonomy (D/C ratio), with a D/C ratio > 1 indicating occupational stress and a D/C ratio ≤ 1 indicating no occupational tension;Sleep quality: Assess the sleep quality of steelworkers according to the internationally accepted Athens insomnia scale [20] (AIS), which is divided into no sleep disorder (overall score ≤ 6) and insomnia (total score > 6) according to the score;DASH score: The DASH dietary model (Dietary Approaches to Stop Hypertension) encourages the intake of five major food groups (fruits, vegetables, nuts and legumes, low-fat milk, and whole grains) to be positively scored; the higher the frequency of intake, the higher the score. The three major food groups restricted by the DASH model (sodium-containing foods, red and processed meats, and sweetened beverages) were negatively scored;r the more frequently they were consumed, the lower the score.

### 2.5. Sample Size Calculation

The sample size calculation method for developing a clinical prediction model proposed by Richard et al. was used [21].

To ensure that the model could accurately predict the mean of the outcome events, the prevalence of hyperuricemia ɵ approximately 12% [22] was reviewed in the literature, and the margin of error δ was set at 0.05, which was calculated to require at least 144 study subjects.
(1)$$n={\left(\frac{1.96}{\delta }\right)}^{2}\;\theta \left(1-\theta \right)$$

In order to control the minimum mean error of all individual prediction values, the mean absolute error MAPE was set to 0.05, the expected shrinkage rate R_CS_^2^ was set to 0.1, and the predictor variable P was about 15, which was calculated to require at least 433 study subjects.
(2)$$n=\exp\left(\frac{-0.508+0.259\ln\left(\theta \right)+0.504\ln\left(p\right)-\ln\left(\mathrm{MAPE}\right)}{0.544}\right)$$

To ensure that the expected shrinkage rate was 10% and reduce model overfitting, S was 0.9, the study variable P was about 15, and it was calculated that at least 1274 cases of study subjects were required.
(3)$$n=\frac{p}{\left(s-1\right)\ln\left(1-\frac{R_{\mathrm{CS}}^{2}}{s}\right)}$$

To ensure that the difference between the developed model and R_CS_^2^ optimization adjustment value was minimized, R_CS_^2^ in Equation (4) was 0.1, maxR_CS_^2^ was 0.48, and S was calculated to be 0.81, which was calculated to require at least 600 study subjects.
(4)$$s=\frac{R_{\mathrm{CS}}^{2}}{R_{\mathrm{CS}+{\delta \mathrm{maxR}}_{\mathrm{CS}}^{2}}^{2}}$$
(5)$$n=\frac{p}{\left(s-1\right)\ln\left(1-\frac{R_{\mathrm{CS}}^{2}}{S}\right)}$$

It was calculated that at least 1274 cases were needed to establish the model sample, and a total of 2992 cases were included in this study. The sample size met the needs of the study.

### 2.6. Model Building

The current study consisted of two main phases: (1) variable screening and (2) model development. We used LASSO regression for variable selection, and we screened the significant variables among 54 clinical characteristics by compressing the coefficients to achieve the effect of variable screening. The code for the LASSO regression implementation is shown in Appendix A. Logistic regression models, CNN models, and XG Boost models were then developed based on the selected variables and literature review.

#### 2.6.1. Logistic Regression Model

The logistic regression model was built using the Sklearn package of python 3.6. The code for the logistic model implementation is shown in Appendix A.

#### 2.6.2. CNN Model

The CNN model was built using the Numpy package, and the sigmoid function was used as the excitation function. The code for the CNN model implementation is shown in Appendix A.

#### 2.6.3. XG Boost Model

The XG Boost model was built using the Sklearn package, using the sigmoid function as the excitation function and the BCE (Binary Cross Entropy) binary cross entropy as the loss function. The code for the XG Boost model implementation is shown in Appendix A.

### 2.7. Model Evaluation

The prediction effectiveness of the model was evaluated in terms of discrimination, calibration, and clinical applicability. The discrimination index included sensitivity, specificity, Youden index, ROC curve, and area under the curve. The calibration index includes the Brier score, Log loss, and calibration curve. Clinical applicability was evaluated by DCA graphs.

### 2.8. Statistical Analysis

An Excel 2010 database was established based on the questionnaire and physical examination data to screen the risk factors for HUA in steelworkers, and a prediction model was established based on the screened variables. Count data were described as rates or composition ratios, and the χ^2^ test was used for comparison between groups; ordinal data were described as rates or composition ratios, and the Kruskal–Wallis test was used for comparison between groups. SPSS 26.0 and Python 3.9 statistical software were used. The test level α was set at 0.05, and both two-sided tests were used.

### 2.9. Quality Control

Design phase: review the literature and consult experts to modify and improve the subject scheme; data collection stage: investigators were uniformly trained. Double-checking of data entry was used, and manual and computerized checking of data entry and logical error checking were performed to ensure the authenticity of the data; Data analysis stage: randomly selected training set and test set.

## 3. Results

### 3.1. Study Population

A total of 4518 steelworkers participated in the occupational health screening, removing 989 HUA patients, 385 missed visits, and 152 incomplete information from the baseline survey, resulting in a final cohort size of 2992. The study population was randomly divided into a training cohort (2094) and a validation cohort (898) in a ratio of 7:3, as shown in Figure 1.

### 3.2. Analysis of Study Population Characteristics

The cohort was followed up from March to September 2019 with a median follow-up time of 26 months and 465 new HUA patients and a crude incidence rate of 15.5%, including 16.31% in men and 7.58% in women. A comparative analysis of the basic characteristics of the workers in the training and validation cohorts revealed no statistically significant differences in the indicators, as detailed in Table 1.

### 3.3. Variable Screening

Predictor variables were screened by LASSO regression, and 6 predictor variables were finally screened out of 54 variables, including total cholesterol, BMI, blood pressure, waist circumference, creatinine, and DASH score, as shown in Figure 2. The figure on the left was the LASSO coefficient path diagram, where each curve represents the trajectory of the coefficient of each variable, and the variables first attributed to point 0 were excluded. The figure on the right is the cross-validation curve. The mistakes were the smallest when the parameters corresponding to the dashed line were selected, and the intersection of the dashed line and the abscissa coordinates corresponded to the Lambda in the left figure. Finally, six indicators with a large impact on the study outcome were screened. Throughout the literature review, we found that smoking, alcohol consumption, and physical activity were also important influencing factors of HUA [23], so they were added together to the subsequent model development.

### 3.4. Multicollinearity Test

The predictor variables were tested for multicollinearity, and we found that the variance inflation factors of all variables were greater than 0 and less than 1.4, and the tolerances were between 0 and 1. There was no multicollinearity problem, as shown in Table 2.

### 3.5. Evaluation of Model Effectiveness

The results of the training set of 2094 cases (70%) showed that the XG Boost model was better than the other two models in terms of sensitivity, specificity, Youden index, F1 score, AUC (95% CI), Brier score, and Log loss, respectively. The CNN model had a higher classification accuracy of 86.8%. The Logistic regression model indicators were slightly worse, as shown in Table 3, ROC curves as shown in Figure 3a.

The results of the validation set of 898 (30%) showed that the XG Boost model outperformed the other two models in terms of classification accuracy, sensitivity, specificity, Youden index, F1 score, AUC (95% CI), Brier score, and Log loss, respectively, as shown in Table 3, ROC curves as shown in Figure 3b.

The XG Boost model outperformed both the CNN and Logistic regression models in terms of Brier Score and Log loss, the calibration curves for both the training and validation sets were close to the diagonal, with no serious deviation from the results. Moreover, the XG Boost model performed best in terms of calibration accuracy, with the Logistic Regression model coming second and the CNN model deviating more from the diagonal, as shown in Figure 4.

The clinical decision curves for the three models are shown in Figure 5, among which the XG Boost model had the highest clinical applicability, and the logistic regression and CNN models had slightly worse clinical applicability. The nomogram of HUA risk in steelworkers was shown in Figure 6.

## 4. Discussion

In this study, we used LASSO regression for the screening of predictor variables, and eventually screened out 6 influencing factors among 54 predictor variables. LASSO regression was an advanced variable selection algorithm for high-dimensional data, and the complexity of the model can be simplified by constructing a penalty function to complete the screening of predictor variables [24]. Compared with the traditional stepwise regression method, LASSO regression can simultaneously process all independent variables at the same time, which not only effectively controlled model overfitting, but also made the model much more stable. On top of this, we added three influencing factors of HUA among steelworkers, such as smoking, alcohol consumption, and physical activity, found through the literature review, to improve the efficiency of the study. By comparing the predictive effects of the three different models, we found that the XG Boost model was the optimal model in this study and that the XG Boost model achieved better results in three areas: discrimination (AUROC 0.806), calibration (Brier Score 0.095), and clinical applicability. Our study highlighted the value of occupational health screening data in predicting HUA, and the screening of predictor variables may provide a scientific basis for the prevention and treatment of HUA in steelworkers.

The current study showed that overweight and obesity were important risk factors for the development of HUA in steelworkers, which is similar to the findings obtained from several previous studies [25,26,27]. Some studies have shown that obesity and the development of HUA were causally related to each other and were closely associated with unhealthy dietary habits, alcohol intake, and a sedentary lifestyle [11]. On the one hand, obese people tend to eat more meat, leading to increased exogenous purine intake and causing HUA. On the other hand, obese people ingest more energy than they consume, resulting in hyper-synthesis of purines in the body, leading to increased endogenous uric acid production [28]. An analysis of the US population found that BMI was the most important modifiable risk factor for HUA, with 44% of the population attributing HUA to overweight or obesity [29]. Both previous and current studies suggested that controlling overweight and obesity was beneficial in reducing the incidence of HUA. Dietary factors were also another important factor influencing the occurrence of HUA. The DASH dietary pattern involved in the current study was originally designed and developed to control hypertension and was a dietary pattern focusing on plant-based foods and high-quality protein that not only significantly reduced blood pressure but had also been used for cardiovascular disease prevention. Regarding the effect of the DASH diet on the risk of gout, Sharan conducted a cohort study that included more than 40,000 study subjects with up to 26 years of follow-up, and their results showed that the DASH diet score was negatively associated with the risk of gout [30]. The possible mechanism for this was that the DASH diet was lower in purines, reducing the purine load in the body. In addition, the DASH diet may act by improving insulin resistance in order to reduce SUA levels [31]. The above study supported the view of the current study that the DASH dietary pattern was a protective factor for the occurrence of HUA in steelworkers. Eating more fruits and vegetables and controlling sugar intake can contribute to the primary prevention of HUA in steelworkers.

Some studies have shown that reducing smoking and alcohol consumption, and a less sedentary lifestyle, can contribute to the prevention of HUA [23]. Smoking or second-hand smoke can increase the risk of HUA and gout. The possible reason for this is that smoking can excite the autonomic nervous system and affect the metabolism of purines in the body, with the potential effect of elevating SUA. In addition, the harmful substances in tobacco can adversely affect the respiratory and circulatory systems, leading to slower blood circulation and impaired uric acid excretion [32]. Alcohol consumption was another important influencing factor in the development of HUA. Firstly, the metabolic process of ethanol in the body consumed a large amount of water, which made the SUA value high. Secondly, the metabolism of ethanol was very likely to produce lactic acid, which was excreted through the kidneys and prevents the normal excretion of uric acid [33]. A sedentary lifestyle could lead to increased uric acid due to slower blood circulation. Moderate exercise accelerates metabolism and facilitates the excretion of uric acid. Long-term moderate-intensity aerobic exercise and aerobic exercise combined with strength training could reduce SUA concentrations in HUA patients [34], and there is a positive correlation between the amount of exercise and the decrease in SUA when exercise is performed at aerobic intensity. The possible mechanism is that long-term moderate-intensity aerobic exercise may protect renal function by alleviating the inflammatory response and ameliorating renal injury through pro-uric acid-excretory protein expression. Exercise plays a direct or indirect role in reducing SUA [35].

In this study on the prediction of HUA onset in steelworkers, the XG Boost model achieved better results and was more suitable for the prediction of HUA onset risk in steelworkers. XG Boost is a classification supervision model based on multiple trees, which essentially took the sum of the predicted values of each tree as the final predicted value. XG Boost had excellent computational efficiency, predictive generalization ability, and overfitting control, making it a long-term dominant data science competition solution. Rajdeep used six different machine learning algorithms to predict obesity risk and achieved a classification accuracy of up to 97.87% for the XG Boost model [36]. Savitesh predicted the risk of pre-diabetes in children and adolescents and found that XG Boost was the best classification model with a 10-fold cross-validation score of up to 90.13%. Savitesh integrated the XG Boost algorithm into a screening tool for completing the automatic prediction of pre-diabetes [37]. Shoukun performed miner fatigue identification based on physiological indicators from ECG and EMG and found that the XG Boost model had the best accuracy and robustness with a recognition accuracy of 89.47% and AUC of 0.90, the recognition of miner fatigue based on the XG Boost model is feasible [38]. The unique ability of logistic regression to correct different prevalence rates made it widely used in medical research, but it showed the poor ability of correct classification and low sensitivity in this study. Although the classification ability of the CNN model was relatively strong, it performed poorly in calibration, possibly because it was better at dealing with image problems.

Our study has several advantages. Firstly, in the process of evaluating the sample size, we did not choose the empirical-based estimation algorithm of 10-fold EPV but used the calculation method proposed by Richard that guaranteed the expected shrinkage rate and controls the error of individual prediction values [21], which made the calculation of the sample size of the HUA onset risk prediction model for steelworkers more rigorous. Second, instead of the traditional stepwise regression method, we chose LASSO regression, which allowed for extensive variable screening in the selection of predictor variables. LASSO regression compensated for the shortcomings of stepwise regression in terms of local optimal estimation and effectively helped us in the selection of predictor variables. In addition, we added three recognized influences such as smoking, alcohol consumption, and physical activity, based on the literature review, making the development of a predictive model for HUA in steelworkers of public health significance. Third, during the development of the model, we made a comprehensive determination in terms of discrimination, calibration, and clinical applicability. Fourthly, we developed a nomogram to predict the risk of HUA in steelworkers. The nomogram was clear and intuitive. From the perspective of steelworkers, the nomogram could predict their own risk of developing HUA in the future, and from the perspective of clinicians, the nomogram could be used to quickly and accurately identify workers at high risk of HUA for targeted health education. By understanding their own risk of HUA and raising awareness of risk factors, steelworkers can change their unhealthy lifestyles accordingly and reduce the risk of illness.

Our study has certain limitations. Firstly, as data are not easily available, our study did not find a suitable dataset to externally validate the newly developed HUA risk prediction model for steelworkers. Secondly, we only used traditional machine learning algorithms and did not improve on the relevant algorithms. Therefore, in the future, we will further investigate new algorithms to improve the predictive performance of the model.

## 5. Conclusions

The prediction effect of the XG Boost model was better than the CNN and Logistic regression models and was suitable for the prediction of HUA onset risk in steelworkers.

## Figures and Tables

**Figure 1 ijerph-20-03411-f001:**
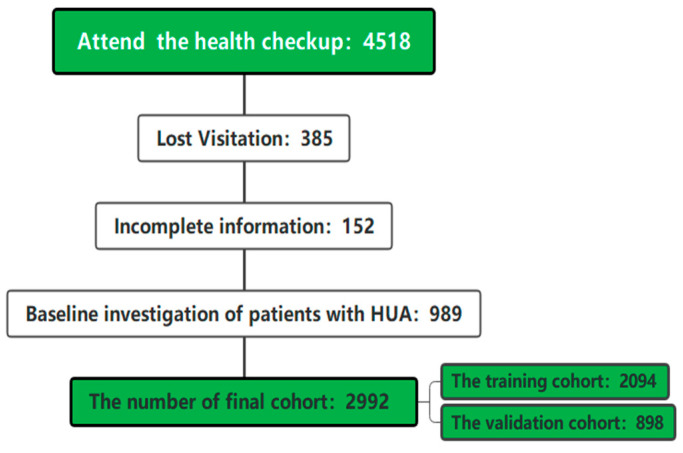
Overview of the study.

**Figure 2 ijerph-20-03411-f002:**
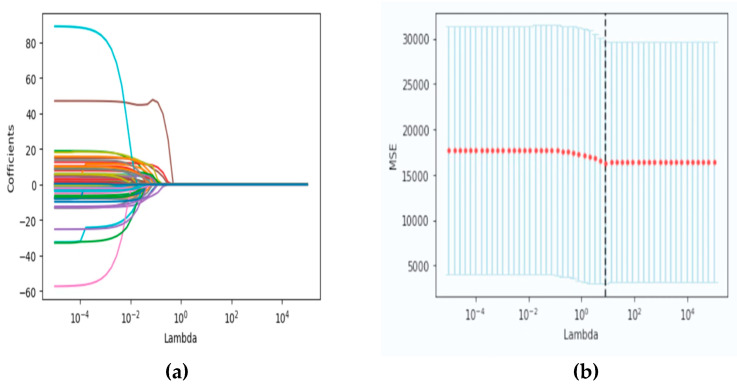
Variable selection based on LASSO regression. (**a**) LASSO coefficient path map; (**b**) Cross validation curve.

**Figure 3 ijerph-20-03411-f003:**
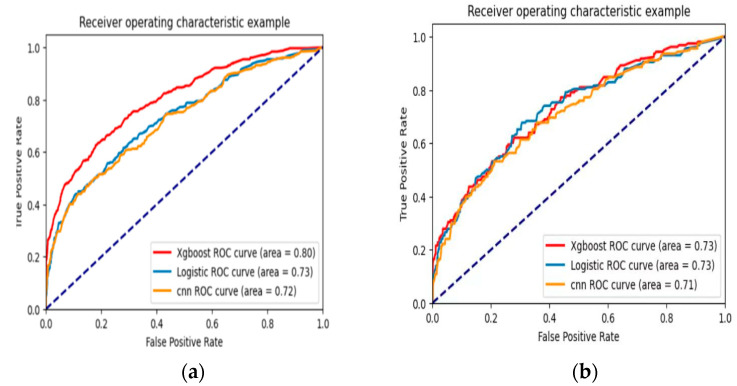
ROC curves of three models. (**a**) Training set; (**b**) validation set.

**Figure 4 ijerph-20-03411-f004:**
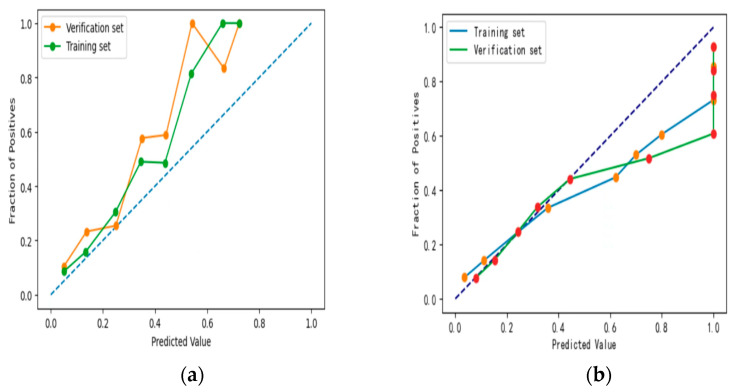
Calibration curves of three models: (**a**) Logistic; (**b**) CNN; (**c**) XG Boost.

**Figure 5 ijerph-20-03411-f005:**
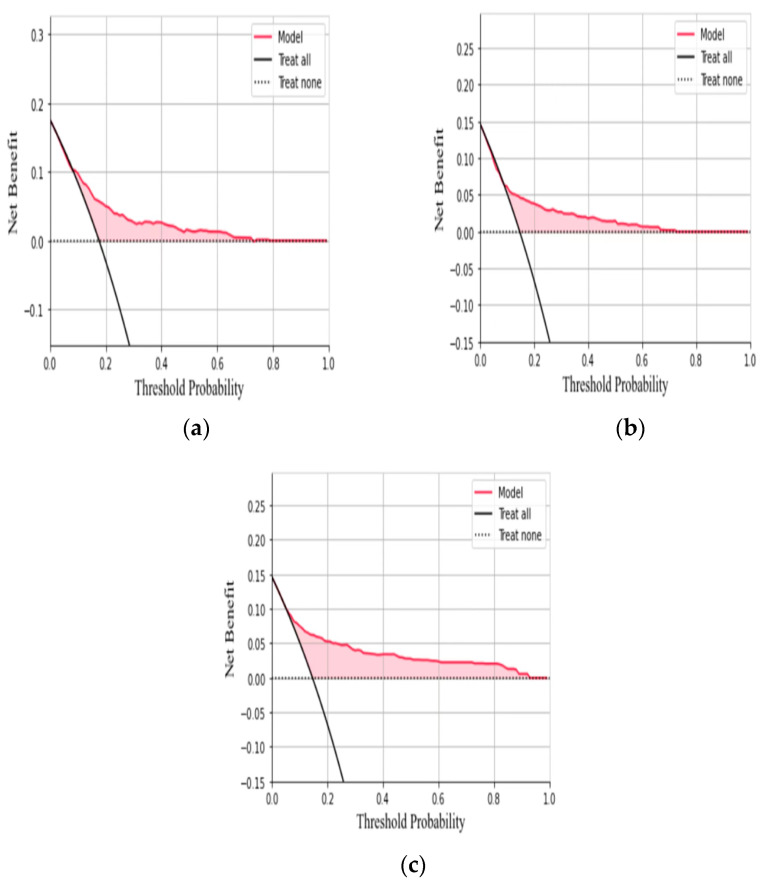
DCA curves of three models: (**a**) Logistic; (**b**) CNN; (**c**) XG Boost.

**Figure 6 ijerph-20-03411-f006:**
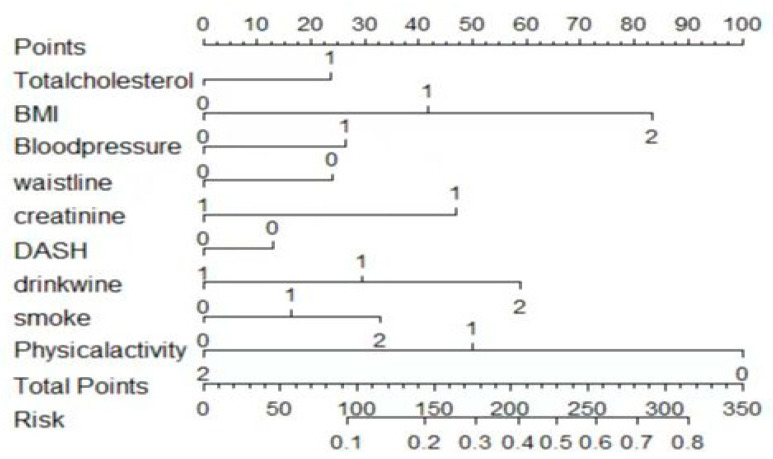
Nomogram of HUA prediction.

**Table 1 ijerph-20-03411-t001:** Characteristics analysis of steelworkers.

Variable	Training Set (2094)		Validation Set (898)		χ^2^/H(K)	*p*
	Total	Patients (%)	Total	Patients (%)		
Age (Year)					0.072 *	0.965 *
<40	403	82 (20.60)	190	39 20.53)		
40~	856	112 (13.08)	369	52 (14.09)		
≥50	838	124 (14.80)	339	56 (16.52)		
Gender					1.676	0.196
male	1930	310 (16.06)	798	135 (16.92)		
female	164	8 (4.88)	100	12 (12)		
Marital status					0.625	0.732
unmarried	43	6 (13.95)	20	4 (20)		
married	1942	275 (14.16)	816	121 (14.83)		
other	109	37 (33.94)	62	22 (35.48)		
Education level					4.507 *	0.105 *
junior secondary school or lower	15	2 (13.33)	6	2 (33.33)		
high school and secondary school	1771	252 (14.23)	775	118 (15.23)		
college and above	308	64 (20.78)	117	27 (23.08)		
Monthly income per capita of the household (Yuan)					0.042 *	0.979 *
<1500	596	64 (10.74)	278	31 (11.15)		
1500~	844	141 (16.71)	359	68 (18.94)		
≥2500	654	113 (17.28)	261	48 (18.39)		
DASH					0.077	0.782
<25	1024	197 (19.24)	434	94 (21.66)		
≥25	1070	121 (11.31)	464	53 (11.42)		
AIS					0.026	0.873
≤6	393	50 (12.72)	151	22 (14.57)		
>6	1701	268 (15.76)	747	125 (16.73)		
Smoking index					0.088 *	0.957 *
0	1024	116 (11.33)	432	49 (11.34)		
<300	500	94 (18.80)	222	45 (20.27)		
≥300	570	108 (18.95)	244	53 (21.72)		
Drinking index					0.153 *	0.926 *
0	1390	156 (11.22)	607	69 (11.37)		
<1028.57	336	81 (24.11)	144	42 (29.17)		
≥1028.57	368	81 (22.01)	147	36 (24.49)		
IPAQ					0.400 *	0.819 *
low	230	82 (35.65)	117	46 (39.32)		
medium	70	19 (27.14)	37	13 (35.14)		
high	1794	217 (12.10)	744	88 (11.83)		
BMI (kg/m^2^)					0.371 *	0.831 *
<24	786	63 (8.02)	356	37 (10.39)		
24~	915	153 (16.72)	389	74 (19.02)		
≥28	393	102 (25.95)	153	36 (23.53)		
Systolic pressure (mmHg)					1.153 *	0.562 *
<120	1002	104 (10.38)	445	60 (13.48)		
120~	839	147 (17.52)	363	69 (19.01)		
≥140	253	67 (26.48)	89	18 (20.22)		
Diastolic pressure (mmHg)					0.905 *	0.636 *
<80	481	52 (10.81)	234	26 (11.11)		
80~	1265	195 (15.42)	534	100 (18.73)		
≥90	348	71 (20.40)	130	21 (16.15)		
Waistline (cm)					0.363	0.547
normal	1204	186 (15.45)	522	74 (14.18)		
abnormal	890	132 (14.83)	376	73 (19.41)		
TG (mmol/L)					0.410	0.522
normal	1829	266 (14.54)	809	123 (15.20)		
abnormal	265	52 (19.62)	89	24 (26.97)		
TC (mmol/L)					0.383	0.536
normal	1866	271 (14.52)	808	122 (15.10)		
abnormal	228	47 (20.61)	90	25 (27.78)		
FPG (mmol/L)					0.489 *	0.783 *
<6.1	1389	201 (14.47)	617	91 (14.75)		
6.1~	520	90 (17.31)	208	40 (19.23)		
≥7.0	185	27 (14.59)	73	16 (21.92)		
HDL-C (mmol/L)					0.071	0.790
≥1.0	1821	267 (14.66)	796	124 (15.58)		
<1.0	273	51 (18.68)	102	23 (22.55)		
LDL-C (mmol/L)					0.081	0.777
<4.1	1826	270 (14.79)	794	124 (15.62)		
≥4.1	268	48 (17.91)	104	23 (22.12)		
Cr (U/L)					0.038	0.845
low	1001	119 (11.89)	441	62 (14.06)		
high	1093	199 (18.21)	457	85 (18.60)		
BUN (mmol/L)					0.468	0.494
<7.1	1974	301 (15.25)	843	141 (16.73)		
≥7.1	120	17 (14.17)	55	6 (10.91)		
Seniority (Year)					0.019*	0.990*
<15	327	31 (9.48)	162	16 (9.88)		
15~	1332	208 (15.62)	574	101 (17.60)		
≥30	435	79 (18.16)	162	30 (18.52)		
JCQ					0.055	0.814
no	750	117 (15.60)	341	61 (17.89)		
yes	1334	201 (15.07)	557	86 (15.44)		
Cumulative noise exposure (dB (A)·Year)					0.146 *	0.930 *
0	921	136 (14.77)	393	68 (17.30)		
<21.34	594	101 (17.00)	244	40 (16.39)		
≥21.34	579	81 (13.99)	261	39 (14.94)		
Dust cumulative exposure (mg/m^3^·Year)					0.128 *	0.938 *
0	1250	193 (15.44)	535	84 (15.70)		
<1374.3	429	52 (12.12)	172	26 (15.12)		
≥1374.3	415	73 (17.59)	191	37 (19.37)		
Cumulative exposure to high temperatures (°C·Year)					0.321 *	0.852 *
0	1211	134 (11.07)	462	63 (13.64)		
<568.5	500	98 (19.60)	203	39 (19.21)		
≥568.5	473	86 (18.18)	233	45 (19.31)		
Accrued days for night shifts (Day)					1.160 *	0.560 *
0	278	31 (11.15)	118	21 (17.80)		
<1976.4	915	156 (17.05)	387	60 (15.50)		
≥1976.4	901	131 (14.54)	397	66 (16.62)		

* Ordinal data were compared between groups using the Kruskal-Wallis test.

**Table 2 ijerph-20-03411-t002:** Multicollinearity test of predictor variables.

Variable	Tolerance	VIF
TC	0.986	1.014
BMI	0.760	1.316
Hypertension	0.825	1.212
Waistline	0.779	1.284
Cr	0.996	1.004
DASH	0.901	1.110
Smoking index	0.897	1.114
Drinking index	0.898	1.114
IPAQ	0.894	1.119

**Table 3 ijerph-20-03411-t003:** Performance of the three models.

Evaluation Indicator	Logistic		CNN		XG Boost	
	Training Set	Validation Set	Training Set	Validation Set	Training Set	Validation Set
Accuracy (%)	84.4	83.9	86.8	85.0	86.6	88.1
Sensitivity (%)	68.4	66.0	72.3	70.5	81.5	78.1
Specificity (%)	82.0	79.7	85.7	80.6	86.8	84.6
Youden index	0.504	0.457	0.580	0.511	0.683	0.627
F1 Score	0.239	0.208	0.207	0.140	0.343	0.278
AUC (95% *CI*)	0.734 (0.685, 0.782)	0.731 (0.676, 0.785)	0.724 (0.669, 0.779)	0.713 (0.667, 0.759)	0.806 (0.748, 0.863)	0.733 (0.689, 0.777)
Brier Score	0.121	0.126	0.194	0.122	0.095	0.107
Log loss	0.398	0.413	0.442	0.409	0.328	0.361

## Data Availability

Data are available on request due to restrictions privacy. The data presented in this study are available on request from the corresponding author. The data are not publicly available due to the data being not readily available.

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
