# Peer review of "Risk Prediction for the Development of Hyperuricemia: Model Development Using an Occupational Health Examination Dataset"

_ijerph, 2023, doi:10.3390/ijerph20043411_

Round 1

Reviewer 1 Report

Overall, the methods and results of the paper are well presented. Only two minor problems:

1. Line 190, 193: the author should attach the supplementary materials;

2. Figure 2: the author should indicate the six predictor variables in the graph to show the significance. Also, figure 2 has two graphs, the author should discuss them separately and have a figure legend for each panel.

Author Response

Dear reviewer
        We appreciate your comments on our paper. We have tried our best to revise the manuscript according to your comments. Thank you again for your hard work and look forward to your reply.

Best regards

Reviewer 2 Report

The manuscript, entitled ‘Risk prediction for the development of hyperuricemia: model development using an occupational health examination dataset’ provides an interesting mathematical model to define different risks and the subsequent development of hyperuricemia in steel workers

The reviewer considers the proposed work to be an excellent example of interdisciplinary work aimed at the deep description of critical issues related to environmental quality and public health. Therefore, by addressing in a technical manner a problem related to socio-environmental management that affects quality of life, the reviewer believes that it perfectly fulfils the aims of the journal

However, as the journal does not specialize in static and/or mathematical matters, the authors should better clarify the mathematical models used and the deductions they draw from model comparisons, to make the reading more usable for less experienced readers. 

All proposed figures, graphical contents, should be better represented and explained also in their captions

Author Response

(The authors gave the same response as above.)

Reviewer 3 Report

Studies comparing different classification methods are of great interest. This study is also interesting and the application to HUA has clinical relevance. The conclusion that a typical AI method line CNN is not the best is also important for "data scientists" who often resort to such methods.

However, there are several points that may need clarification.

1.  Most important is an explanation/interpretation why these methods differ in their performance. How generalizable is this finding?

2, Lasso regression is used but not what type. Lasso logistic regression? If so why wasn't this method included instead of standards Logistic Regression? 

3. It would be nice for a wider audience to include a few words/sentences about the underlying models, e.g. that XGboost is a tree based method.

4. It may be worth including a statement (caveat) that logistic regression is easily calibrated to different populations with different prevalences by adjusting then model intercept (provided odds ratios are robust). The other two methods do not have this advantage.

Author Response

(The authors gave the same response as above.)
